# Impact of HLA Polymorphism on the Immune Response to Bacillus Anthracis Protective Antigen in Vaccination versus Natural Infection

**DOI:** 10.3390/vaccines10101571

**Published:** 2022-09-20

**Authors:** Stephanie Ascough, Rebecca J. Ingram, Karen K. Y. Chu, Stephen J. Moore, Theresa Gallagher, Hugh Dyson, Mehmet Doganay, Gökhan Metan, Yusuf Ozkul, Les Baillie, E. Diane Williamson, John H. Robinson, Bernard Maillere, Rosemary J. Boyton, Daniel M. Altmann

**Affiliations:** 1Faculty of Medicine, Imperial College, London W12 0NN, UK; 2Wellcome-Wolfson Institute of Experimental Medicine, Queen’s University Belfast, Belfast BT7 1NN, UK; 3Sanofi, South San Francisco, CA 94080, USA; 4BioMET, School of Medicine, University of Maryland, Baltimore, MD 21201, USA; 5Defence Science Technology Laboratory, Porton Down, Salisbury SP4 0JQ, UK; 6Department of Medical Genetics, Erciyes University Hospital, Kayseri 38095, Turkey; 7Department of Infectious Diseases and Clinical Microbiology, Hacettepe University Faculty of Medicine Ankara, Ankara 06000, Turkey; 8School of Pharmacy and Pharmaceutical Sciences, Cardiff University, Cardiff CF24 4HQ, UK; 9Institute for Cellular Medicine, Newcastle University, Newcastle upon Tyne NE1 7RU, UK; 10CEA-Saclay, Département Médicaments et Technologies pour la Santé, Université Paris-Saclay, 91192 Gif-sur-Yvette, France; 11Lung Division, Royal Brompton and Harefield Hospitals, Guy’s and St Thomas’ NHS Foundation Trust, London SE1 9RT, UK

**Keywords:** anthrax, protective antigen, HLA class II, HLA transgenic, CD4 epitope, HLA-binding, bacterial immunity

## Abstract

The causative agent of anthrax, Bacillus anthracis, evades the host immune response and establishes infection through the production of binary exotoxins composed of Protective Antigen (PA) and one of two subunits, lethal factor (LF) or edema factor (EF). The majority of vaccination strategies have focused upon the antibody response to the PA subunit. We have used a panel of humanised HLA class II transgenic mouse strains to define HLA-DR-restricted and HLA-DQ-restricted CD4+ T cell responses to the immunodominant epitopes of PA. This was correlated with the binding affinities of epitopes to HLA class II molecules, as well as the responses of two human cohorts: individuals vaccinated with the Anthrax Vaccine Precipitated (AVP) vaccine (which contains PA and trace amounts of LF), and patients recovering from cutaneous anthrax infections. The infected and vaccinated cohorts expressing different HLA types were found to make CD4+ T cell responses to multiple and diverse epitopes of PA. The effects of HLA polymorphism were explored using transgenic mouse lines, which demonstrated differential susceptibility, indicating that HLA-DR1 and HLA-DQ8 alleles conferred protective immunity relative to HLA-DR15, HLA-DR4 and HLA-DQ6. The HLA transgenics enabled a reductionist approach, allowing us to better define CD4+ T cell epitopes. Appreciating the effects of HLA polymorphism on the variability of responses to natural infection and vaccination is vital in planning protective strategies against anthrax.

## 1. Introduction

Anthrax is an acute zoonotic disease that primarily affects grazing mammals, although the causative agent, Bacillus anthracis, also infects humans and is found in many parts of the developing world, where the majority of natural human infections occur [1]. Infections in humans, which may be fatal, depending upon the route of infection, are usually confined to agricultural workers, those who eat infected carcasses and those who handle the skins and coats of infected animals [2]. Over past decades, the need to protect individuals from occupational exposure has combined with fears regarding the use of anthrax as a bioweapon, to drive the development of vaccines based on the toxins produced by the bacteria [1]. Such concerns have resurfaced recently in relation to potential anthrax weaponisation [3]. Furthermore, there have been recent cases in Northern Europe of anthrax infections in intravenous drug users as a consequence of contaminated drug supplies [4]. There are also growing concerns regarding the effect of climate change in the Arctic upon the release of potentially viable anthrax spores from melting permafrost [5].

The three toxins of *B. anthracis*, Protective Antigen (PA), Lethal Factor (LF) and Edema Factor (EF) combine in a binary fashion, so that coupling PA with LF or EF produces Lethal Toxin (LT) or Edema Toxin (ET), respectively [6]. The two predominantly used vaccines, the United States-licensed Anthrax Vaccine Adsorbed (AVA; trade name BioThrax) and the United Kingdom-licensed vaccine, Anthrax Vaccine Precipitated (AVP), are culture filtrate vaccines containing PA and variable amounts of LF and EF [7]. Both vaccines are administered intramuscularly: AVA is given as three initial doses at 0, 1 and 6 months, while AVP is administered as a primary series of four vaccinations at 0, 3, 6 and 32 weeks [6]; a booster vaccination at 12 months, after the primary series for each vaccine, is then required. The requirement for an intensive vaccination regimen, as well as concerns about adverse reaction rates as high as 11% for the UK vaccine [8], and up to 60% for the US vaccine [9], have prompted interest in streamlined vaccination schedules or the development of effective, safe, subunit vaccines [10,11]. Recent efforts to simplify the immunisation regimen have resulted in a reduced frequency of boosting after the primary schedule (ten-yearly for AVP and three-yearly for AVA licensed in Europe) [12].

Second-generation anthrax vaccines under development are based on the administration of the immunogenic anthrax toxins, specifically recombinant protective antigen (rPA). Human clinical trials have indicated that these rPA vaccines may be capable of eliciting robust cellular and humoral immune responses, whilst avoiding the adverse reactions associated with older filtrate-based vaccines [13,14,15].

PA-specific monoclonal antibodies generated from AVA-vaccinated humans were found to neutralise LT in vitro, and passive transfer of these antibodies provided protection in mouse models of LT challenge [16,17]. Although it is possible to show passive transfer of immunity with toxin-neutralising antibodies [18], Crowe et al. found that over half of AVA-vaccinated individuals demonstrated no detectable toxin-neutralising effect; despite the presence of anti-PA antibodies in the majority of vaccinated individuals [19]. Studies in rhesus macaques have demonstrated that AVA administration is capable of providing protection from subsequent spore challenge, with a Th1/Th2 profile predictive of survival, even in the presence of very low levels of circulating anti-PA antibody [20].

Protection afforded by a response to PA in both rodent and non-human primate models has been suggested to be T-cell mediated [21,22]. Plasmid vaccination in mice induces high antibody titres as well as PA-specific Th1 immunity and induction of a high level of IFNγ secretion [23]. Doolan and colleagues reported that individuals exposed to anthrax spores in the US mail service incident experienced dose-dependent priming of T cell immunity, and, to a lesser extent, of B cell immunity against PA [24]; low-level anthrax exposure led to PA T cell responses in the absence of detectable antibodies. While Glomski et al. found that, in contrast to humoral immunity, IFNγ production by CD4+ T cells protected mice against capsulated *B. anthracis* infection [25].

Recent advances in vaccinology and microbial immunology have indicated that there is strong evidence for HLA polymorphisms as a determinant of susceptibility and resistance to disease, and previous work from our lab has shown that individuals naturally exposed to anthrax spores demonstrate IFNγ secreting antigen-specific CD4+ T cell immunity to PA and LF, which, for PA, showed correlation between the magnitude of response and the duration of the infection [26,27]. We also found that a survivor of injectional anthrax developed strong, potentially protective, T cell immunity to several commonly immunodominant epitopes of PA and LF, previously described in Turkish patients [28]. This evidence suggests that cellular immunity, especially mediated by antigen presentation through both HLA-DR and DQ alleles to a specific TCR repertoire, has a critical role to play in vaccine mediated clearance of *B. anthracis*.

Whether the future of anthrax vaccinology lies with third-generation, subunit vaccines or with improved protocols for priming with existing vaccines, the need has never been greater to fully comprehend the nature of effective immunity to *B. anthracis*, and the impact of immunogenetic diversity.

The present study aims to characterise CD4+ T cell immunity to the PA toxin derived from *B. anthracis*. This encompasses a comprehensive analysis of T cell epitopes through investigation of HLA class II binding, mapping of responses in a panel of HLA class II transgenic mice, live challenge studies in HLA transgenic mice and studies of infected or vaccinated human donors. Our results show PA to be highly CD4+ T cell epitope-rich, with variable immunodominance which is dependent on HLA class II genotype. As discussed below, this has implications for wide-scale roll-out and assessment of PA-based vaccines.

## 2. Materials and Methods

### 2.1. Ethics Statement

Human blood samples from Kayseri (Turkey) were obtained with full review and approval by The Ethics Committee of the Faculty of Medicine, Erciyes University. Human vaccinees based at DSTL, Porton Down, participated in the context of a study protocol approved by the CBD IEC (Chemical and Biological Defence Independent Ethics Committee). Written informed consent was obtained from all human volunteers who were assessed for enrolment using protocol-defined inclusion and exclusion criteria (Appendix A). Clinical and demographic details for these individuals, including duration and severity of infection, number of vaccinations and time since most recent vaccination, have been previously reported in Ingram et al. JI 2010 [26]. All mouse experiments were performed under the control of UK Home Office legislation, in accordance with the terms of the Project License (70/5994) granted for this work under the Animals (Scientific Procedures) Act 1986, having also received formal approval of the document through the Imperial College Ethical Review Process (ERP) Committee.

### 2.2. HLA Class II Transgenic Mice

HLA class II transgenic mice carrying genomic constructs for HLA-DRA1*0101/HLA-DRB1*0101 (HLA-DR1), HLA-DRA1*0101/HLA-DRB1*0401 (HLA-DR4), HLA-DRA1*0101/HLA-DRB1*1501 (HLA-DR15) and HLA-DQA1*0301-DQB1*0302 (HLA-DQ8), crossed for more than six generations to C57BL/6 H2-Ab00 mice, were generated as described previously, to produce mice lacking expression of endogenous murine MHC class II heterodimers, which differed in expression of specific HLA-DR or HLA-DQ alleles. [29,30,31,32]. All experiments were performed in accordance with the Animals (Scientific Procedures) Act 1986, and were approved by local ethical review panel.

### 2.3. Live B. anthracis Challenge

Preliminary data indicated that there was a divergence in the susceptibility of mouse strains to anthrax challenge. Therefore, naïve mice were challenged with *B. anthracis* STI strain by the intraperitoneal route at one of two dose levels: 11 HLA-DR1 and 10 HLA-DQ8 mice were challenged with 10^6^ colony forming units (CFU), while 9 HLA-DR15, 10 HLA-DR4, 8 HLA-DQ6 and 10 C57Bl6 were challenged with 10^4^ CFU per mouse. The animals were monitored for 20 days post-infection, after which all survivors were sacrificed and their spleens were removed and homogenised in 1 mL of PBS, before plating out onto L-agar plates. Colonies were counted after 24 h of culture at 37 °C, and the mean bacterial count per spleen was determined.

### 2.4. Expression and Purification of PA Antigens

Good Manufacturing Practice grade rPA was provided by Avecia Vaccines (Billingham, UK) and had endotoxin levels of <1 EU/mg. Individual domains of PA and peptides were expressed in E. coli and purified as previously described [33]. All proteins and peptides were resuspended in DMSO at 25 mg/mL.

### 2.5. PA Epitope Mapping in Transgenic Mice

Mice were immunised in one hind footpad with 50 μL of 12.5 μg recombinant full-length PA, PA peptides, or a control of PBS, emulsified in an equal volume of TiterMax Gold (Sigma-Aldrich, Burlington, MA, USA) by syringe extrusion. After 10 days, immunised draining popliteal lymph nodes were removed and disaggregated into single-cell suspensions by filtration through 0.7 μm cell strainers. Lymph node cell responses were recalled in vitro with 25 µg/mL of either rPA, truncated PA domains comprising the PA protein, or the overlapping 20-mer peptides covering the full-length PA sequence. This produced a CD4+ T cell epitope map of the entire PA protein sequence. To confirm the immunodominant epitopes identified by this large-scale mapping, mice were then immunised subcutaneously with 12.5 μg of the individual PA peptides in TitreMax adjuvant. After 10 days the lymph node cells were challenged in vitro with 25 µg/mL of the recombinant full-length PA and the immunising and two flanking PA peptides. Quantification of murine antigen-specific INFγ levels was carried out by ELISpot (Besancon Cedex, France) analysis of T cell populations directly ex vivo. Ninety-six-well hydrophobic polyvinylidene difluoride membrane-bottomed plates (MAIP S 45; Burlington, MA, USA) were pre-wetted with 70% ethanol. The plates were washed twice with PBS, then coated with anti-INFγ monoclonal antibody at 4 °C overnight. After blocking with 2% skimmed milk, plates were washed with PBS, and 100 μL/well of antigen was added in triplicate to appropriate wells. For each assay, a medium-only negative control and a positive control of staphylococcal enterotoxin B (SEB 25 ng/mL) were included. Wells were seeded with 2 × 10^6^ cells/mL in HL-1 medium (supplemented with 1% L-glutamine, 1% penicillin/streptomycin, and 2.5% β-mercaptoethanol) and plates were incubated for 72 h at 37 °C with 5% CO_2_. The plate contents were then discarded and plates were incubated with PBS/Tween 20 (0.1%) for 10 min at 4 °C. Plates were then washed twice with PBS/Tween 20 (0.1%) and incubated with biotinylated anti-INFγ monoclonal antibody. Plates were again washed twice with PBS/Tween 20 (0.1%), and then incubated with streptavidin-alkaline phosphatase conjugate. After a wash with PBS/Tween 20 (0.1%), plates were treated with 5-bromo-4-chloro-3-indolyl phosphate and nitro blue tetrazolium (BCIP/NBT) and spot formation was monitored visually. The plate contents were then discarded and plates were washed with water, then air-dried and incubated overnight at 4 °C to enhance spot clarity. Spots were counted using an automated ELISpot reader (AID), and results expressed as delta spot-forming cells per 10^6^ cells (ΔSFC/10^6^, which was calculated as SFC/10^6^ of stimulated cells minus SFC/10^6^ of negative control cells). The results were considered positive if the ΔSFC/10^6^ was more than two standard deviations above the negative control. For assessment of peptide-specific T cell proliferation, murine lymphocytes were resuspended at 3.5 × 10^6^ cells/mL in supplemented HL-1 media (Lonza, Slough, UK) (1% L-glutamine, 1% penicillin/streptomycin, 2.5% β-mercaptoethanol) and 100 μL/well was plated out in triplicate in 96-well Costar tissue culture plates (Corning Incorporated, Glendale, AZ, USA). The cells were stimulated with 100 μL/well of appropriate antigen, positive controls of 5 μg/mL Con A (Sigma-Aldrich, USA) or 25 ng/mL of SEB (Sigma-Aldrich, St. Louis, MO, USA) or negative controls of medium with cells. Plates were incubated at 37 °C with 5% CO_2_ for 5 days. Eight hours before harvesting, 1 μCi/well of [3H]-thymidine (GE Healthcare, Chalfont St. Giles, Buckinghamshire, UK) was added. The cells were harvested onto fiberglass filter mats (PerkinElmer, Waltham, MA, USA) using a Harvester 96 cell harvester (Tomtec, Hamden, CT, USA) and counted on a Wallac Betaplate scintillation counter (EG&G Instruments, Netherlands). Results were expressed as either delta counts per minute (ΔCPM, which was calculated as CPM of stimulated cells minus CPM of negative control cells) or stimulation index (SI, which was calculated as CPM of stimulated cells divided by CPM of negative control cells). An SI of ≥2.5 was considered to indicate a positive proliferation response.

### 2.6. PA Epitope Mapping with Human Donor PBMC Samples

Lymphocytes were isolated from human peripheral blood samples and stimulated as described previously [26]. In brief, sodium-heparinised blood was collected with full informed consent (Ericyes University Ethical Committee) from nine Turkish patients treated for cutaneous anthrax infection within the last eight years and 10 volunteers routinely vaccinated every 12 months for a minimum of five years with the UK AVP vaccine Peripheral blood mononuclear cells (PBMC) were isolated from the blood by centrifugation at 800× *g* for 30 min in Accuspin tubes (Sigma, Hertfordshire, UK). Cells were then removed from the interface and washed twice in AIM-V serum free media. Cells were counted for viability and resuspended at 2 × 10^6^ cells/mL. Human T cell INFγ levels were quantified by ELISpot (Diaclone, France), as previously described [26]. In brief, the peptide library was prepared in a matrix comprising six peptides per pool, so that each peptide occurred in two pools but no peptides occurred together in multiple pools. This allowed the determination of responses to individual peptides. The in-well concentration of each peptide was 25 µg/mL and total peptide concentration per well was 150 µg/mL. After addition of antigen to the wells the plates were frozen at −80 °C until use. Wells were seeded with human PBMCs at 2 × 10^5^ cells/well (range: 1.6 × 10^5^ to 2.1 × 10^5^ cells/well) in AIM-V media (Gibco, Waltham, MA, USA) and plates were incubated for 72 h at 37 °C with 5% CO_2_. The plate contents were then discarded and plates were washed with PBS-Tween 20 (0.1%) and incubated with biotinylated anti-INFγ, then washed again before streptavidin-alkaline-phosphatase conjugate was added. After a final wash, plates were developed by addition of BCIP/NBT. Spots were counted using an automated ELISpot reader (AID), and results were expressed as ΔSFC/10^6^. The results were considered positive if the ΔSFC/10^6^ was more than two standard deviations above the negative control and ≥50 spots.

### 2.7. HLA-Peptide Binding Assay

Competitive ELISAs were used to determine the relative binding affinity of PA peptides to HLA-DR molecules, as previously described [34,35]. Briefly, the HLA-DR molecules were immunopurified from homozygous EBV-transformed lymphoblastoid B cell lines by affinity chromatography. The HLA-DR molecules were diluted in HLA binding buffer and incubated for 24 to 72 h with an appropriate biotinylated reporter peptide, and a serial dilution of the competitor PA peptides. Controls of unlabelled reporter peptides were used as reference peptides to assess the validity of each experiment. An amount of 50 μL of HLA binding neutralisation buffer was added to each well and the resulting supernatants were incubated for 2 h at room temperature in ELISA plates (Nunc, Roskilde, Denmark) previously coated with 10 μg/mL of the monoclonal antibody L243. Bound biotinylated peptide was detected by addition of streptavidin-alkaline phosphatase conjugate (GE Healthcare, Buc, France) and 4-methylumbelliferyl phosphate substrate (Sigma-Aldrich, St. Quentin Fallavier Cedex, France). Emitted fluorescence was measured at 450 nm post-excitation at 365 nM on a SpectraMax Gemini fluorometer (Molecular Devices, San Jose, CA, USA). The PA peptide concentration that prevented binding of 50% of the labelled peptide (IC50) was evaluated, and data expressed as relative binding affinity (ratio of IC50 of the PA competitor peptide to the IC50 of the reference peptide that binds strongly to the HLA-DR molecule). Sequences of the reference peptides and their IC50 values were as follows: HA 306–318 (PKYVKQNTLKLAT) for DRB1*0101 (4 nM), DRB1*0401 (8 nM), DRB1*1101 (7 nM), YKL (AAYAAAKAAALAA) for DRB1*0701 (3 nM), A3 152–166 (EAEQLRAYLDGTGVE) for DRB1*1501 (48 nM), MT 2–16 (AKTIAYDEEARRGLE) for DRB1*0301 (100 nM), B1 21–36 (TERVRLVTRHIYNREE) for DRB1*1301 (37 nM), DQB45–57 (ADVEVYRAVTPLGPPD) for DQ8 (100 nM) and INS1–15A (FVNQHLAGSHLVEAL) for DQ6 (100 nM). Good binding affinity was defined in this study as a relative activity <100.

## 3. Results

### 3.1. CD4+ T Cell Responses to *B. anthracis* PA Epitopes in Anthrax-Recovered Patients and Vaccinees

In this study we sought to describe T cell memory responses to anthrax antigens in two cohorts of individuals who either suffered clinical disease after natural, occupational exposure, or were vaccinated with the UK AVP anthrax vaccine. In earlier studies, we described the fact that responses to recombinant PA and LF antigens were higher in naturally exposed individuals than in vaccinees receiving a full course of the UK AVP anthrax vaccine [26,36]. Furthermore, immune responses in naturally infected donors were characterised by a broad cytokine profile, encompassing IL-2, IL-5, IL-9 and IL-13 [37]. In the present study we sought to analyse in greater detail the epitope specificity of vaccinated and infected individuals to PA. PA epitopes were screened by looking for ELISpot responses to a panel of 73 overlapping peptides of 20 mers overlapping by 10 amino acid residues and analysed in pools of six. A total of 26 peptides were identified as epitopes in at least one AVP vaccinee (Figure 1), of which only 7 epitopes were an immune target for more than one vaccinee. Of note is the finding that only 4 vaccinees (AVP vaccinees donors 1–4) out of 10 responded to any epitopes, and, of these, the majority of the responses were elicited in donor 3, who responded to a total of 21 epitopes (Appendix A). Although this study was not empowered to make assumptions regarding the involvement of HLA alleles in the presentation of anthrax peptides, it was interesting that HLA-DR11 and DR13 were over-represented in the population of donors responding to the peptides contained within the vaccine. In contrast, the majority of infected individuals (7 out of 9 donors) responded to at least one PA epitope, and there did not appear to be any particular bias towards specific HLA alleles in the responses (Appendix A), with 69 of the 73 peptides analysed in this cohort found to carry infection-specific epitopes. Peptides such as PA 168–187 and PA 651–670 contained epitopes that were recognised with a high frequency response by multiple individuals (PA 168–187 mean = 264.2 spots/million, ±123.2 SEM, and PA 651–670 mean = 273.4 spots/million, ±123.6 SEM) and encompassed diverse HLA class II alleles. However, it was notable that, although adjacent peptides (PA 161–187 and PA 641–660 respectively) were identified as epitopes for one of the vaccinated individuals, neither of the infection-specific epitopes, recognised in the context of multiple HLA alleles, were a focus of the response in any vaccinees.

In both infected and vaccinated cohorts, the epitopes came from sequences within all four domains of PA (Figure 1), indicating that, unlike LF, the majority of PA epitopes are not clustered within a single domain of the protein [26]. This comparison also highlighted the fact that individuals who had been hyper-immunised on the standard UK schedule with seven to 14 doses of the AVP vaccine over 3.5 to 10 years, responded to fewer epitopes than infected individuals, with no epitopes identified that were present in the context of vaccination alone. This supported the suggestion, which we originally made in regard to LF, that live infection unveils cryptic anthrax epitopes not commonly recognised after administration of the protein antigen.

### 3.2. Differential Susceptibility to *B. anthracis* Challenge in HLA Transgenic Mice

In order to more precisely define the contribution of different HLA class II alleles to anthrax and PA immunity, we turned to HLA class II transgenic mice as a defined, reductionist model, allowing analysis of individual alleles in isolation.

The challenge dose used in this study was based on data gained from two preceding small-scale susceptibility studies, in which it was found that HLA-DR1 and HLA-DQ8 transgenics were resistant to challenge with 1 × 10^5^ CFU, and required a higher challenge dose, relative to the other, more susceptible, HLA strains, to establish infection. We therefore compared susceptibility of mice expressing either HLA-DR1 or HL-DQ8 to challenge with 1 × 10^6^ CFU (10^3^ median lethal doses, MLD) *B. anthracis* STI strain. HLA-DR1 mice were resistant to *B. anthracis* STI challenge (MLD > 10^6^ CFU), while HLA-DQ8 mice were also relatively resistant, with 80% survival. The more susceptible HLA class II transgenic mice demonstrated differential susceptibility to challenge at 10^5^ CFU (10^2^ MLD *B. anthracis* STI) with the following survival rates: DQ6 mice (100%), DR4 (80%), and DR15 (55%). By comparison, the parent strain for the HLA class II transgenics, C57BL6, showed 40% survival against a 10^5^ CFU contemporaneous challenge with the STI vaccine strain of *B. anthracis*.

The bacterial loads recovered from the spleens of individual surviving mice of each strain at day 20 are shown in Figure 2. In general, the mean bacterial loads in spleens at day 20 post-infection were lower than, but proportional to, the original challenge dose level. The survivors of the groups challenged with 10^6^ CFU (DR1, DQ8) had high bacterial loads in their spleens, even at 20 days post infection (Appendix A), although the mean bacterial loads for the DQ6 survivors at the same time point (challenged with 10^5^ CFU) did not differ significantly from those for the DR1 or DQ8 mice. As these two transgenic strains had been challenged with ten-fold more bacteria, this suggested that the DQ6 mice were slower to clear the infection.

HLA transgenic mice were less susceptible to infection with *B. anthracis* STI strain than the parent strain C57BL6 mice. HLA-DR1 mice were resistant to infection with a high-level challenge (10^6^ CFU). DQ6 strain mice were resistant to 10^5^ CFU and relatively slow to clear the infection. The order of susceptibility of mouse strains to *B. anthracis* infection was determined to be: C57Bl6 > DR15 > DR4 > DQ6 > DQ8 > DR1. 

### 3.3. CD4+ T Cell Responses to *B. anthracis* PA Epitopes in HLA Transgenic Mice

The greater immunogenetic complexity of HLA-outbred human populations makes it considerably more challenging to define the restricting HLA molecule responsible for antigen presentation. The HLA class II transgenic mouse models offer a reductionist system in which to define HLA-restricted epitopes of relevance to humans carrying the same alleles. Using these transgenic models in protein and peptide immunisation, we were able to build a comprehensive picture of immunodominant HLA class II restricted epitopes derived from PA. Mice were immunised with the recombinant PA protein and draining lymph node cells were restimulated with a peptide library spanning the PA sequence (73 peptides in total, with some peptides overlapping the boundaries between domains: domain 1 = PA 1–20 to PA 241–260; domain 2 = PA 251–270 to PA 471–490; domain 3 = PA 491–510 to PA 581–600; domain 4 = PA 591–610 to PA 716–735,). After immunisation with the recombinant protein of interest, all HLA transgenic mice responded to the whole rPA (Figure 3), but the response to the individual peptides was found to be HLA-specific.

We investigated whether there might be any correlation between susceptibility of the HLA transgenic lines to challenge and the breadth of T cell epitope recognition. Antigen-specific T cell responses to all stimulatory peptides were further investigated in depth by peptide immunisation and screening (Appendix A). In total, 6 HLA-DR1 restricted epitopes were identified (Figure 3A and Appendix A). In comparison, 14 HLA-DQ8 restricted epitopes were identified (Figure 3B and Appendix A), and 15 HLA-DR4 restricted epitopes were identified (Figure 3C and Appendix A). Whilst some of these epitopes were recognised by more than one HLA type (PA 331–350, PA 591–610, PA 601–620 and PA 711–730 were constituents of both DR4 and DQ8 responses, while PA 371–390 and PA 681–700 were recognised by both DR1 and DR4 alleles), no one epitope was found to provoke a response in all 3 HLA alleles tested.

It was noteworthy that HLA-DR1 transgenic mice, which were the least susceptible to anthrax challenge, responded to fewer epitopes with a reduced repertoire of CD4+ T cell recognition than the other HLA alleles screened. This was not a simple reflection of differences in HLA transgene expression by the different HLA class II transgenic mouse strains or CD4+ positive selection, as HLA-DR4 transgenics present a similar number of epitopes to HLA-DQ8, despite showing the highest level of HLA class II expression [27]. We also previously found a pronounced hierarchy of response in relation to the LF protein, with HLA-DR1 transgenics mounting a considerably larger response than HLA-DQ6, DR15 or DR4 transgenics, and HLA-DR4 transgenics conversely showing the weakest response [27].

### 3.4. The Differential PA Peptide Binding across Distinct HLA Polymorphisms

Overlapping 20-mer peptides that represented the whole PA protein sequence were evaluated for binding affinity to seven common HLA-DR alleles and two common HLA-DQ alleles (Table 1). The two epitopes that were recognised by multiple individuals from the infected cohort (PA 168–187 and PA 651–670) showed a complete disparity in their HLA binding affinities. Whilst PA 168–187 was not recognised by any of the transgenic lines and showed an exceptionally low binding affinity across all HLA-DR alleles tested, PA 651–670 showed strong-to-moderate binding across all HLA-DR alleles, and bound strongly to HLA-DQ8, which also correlated with a strong response seen in the corresponding transgenic line. Overall, we were not able to identify a propensity towards a strong HLA binding affinity in those epitopes that were a feature of the infected response. In contrast, all but one (PA 501–520) of the seven epitopes identified in more than 20% of the vaccinated cohort demonstrated good binding affinities for the HLA-DR or DQ alleles carried by those individuals. This suggested that the binding affinity might be a more important predictor of epitope hierarchy in the context of vaccination than infection.

### 3.5. Figures and Tables

Heat map representation of the epitope mapping results were observed for positive CD4+ T cell IFNγ ELIspot responses in the human donor cohorts, comprising a total of 9 donors in the cutaneous anthrax (Kayseri) group and 10 donors in the AVP vaccinees (UK) group. Peptides were considered positive for the carriage of a CD4+ T cell epitope if the response was >50 SFC/106 PBMCs and 2SD above negative control, and the stimulation index (peptide response/negative control response) value was ≥1.5. The domains were defined as described previously; domain 1 = PA 1–20 to PA 241–260; domain 2 = PA 251–270 to PA 471–490; domain 3 = PA 491–510 to PA 581–600; domain 4 = PA 591–610 to PA 716–735, with some peptides overlapping the boundaries between domains) [38]. The colour bar at the right indicates the percentage of donors responding to a given epitope, with shading from white (0%) to dark blue (50%).

Groups of naïve HLA transgenic (DR1 n = 11, DQ8 n = 10, DR15 n = 9, DR4 n = 10, DQ6 n = 8) or C57Bl6 (n = 10) mice were challenged with either 10^5^ (C and D) or 10^6^ (A and B) CFU *B. anthracis* STI strain, in order to compare susceptibility. Mice were challenged intraperitoneally and their survival observed for 20 days post-infection. Percentage survival, together with mean splenic bacterial counts per HLA type, was shown for mice succumbing within the observation period (days 1 to 19) and for survivors culled at day 20. Statistical comparison of mean bacterial loads by mouse strain (D) indicated that higher bacterial loads were seen in DQ6 in comparison to C57BL/6 (** *p* = 0.0093), DR4 (**** *p* < 0.0001) and DR15 (** *p* = 0.0014), (One-way ANOVA, Tukey’s multiple comparisons).

Groups of HLA transgenic mice were immunised with the whole rPA protein in adjuvant, and the proliferative responses of draining lymph node cells to overlapping synthetic peptides, representing the complete PA sequence, were determined. Scatter plots showed responses of individual mice transgenic for (A) HLA-DR1 (n = 3 for each peptide data point, and n = 11 for the rPA data point), (B) HLA-DQ8 (n = 6 for each peptide data point, n = 18 for the rPA data point) and (C) HLA-DR4 (n = 6 for each peptide data point, and n = 17 for the rPA data point). Data are presented as the SI calculated as the mean CPM of triplicate wells in the presence of peptide divided by the mean CPM in the absence of antigen. Values twice the mean CPM in the absence of antigen were considered positive responses. Confirmed epitopes are highlighted in red.

## 4. Discussion

Human exposure to anthrax spores continues to be of considerable concern in diverse spheres of clinical infectious disease; most commonly, exposure may occur naturally, either after ingestion of infected animals or through contact with infected animal products. Other routes of exposure could occur through deliberate release, acts of bioterrorism, or injection of contaminated drugs by intravenous drug users [3,28,39]; in these contexts, especially the threat of bioterrorist use, there has long been a perceived need to have an effective anthrax vaccination programme available. Three major vaccines have been in use in various parts of the world since the Cold War, with various recombinant subunit vaccine candidates in trial for rollout [40,41]. Interestingly, however, compared to many other bacterial pathogens, the immunology and immunogenetics underpinning any clear understanding of correlates of protection (CoP) are poorly delineated for anthrax [42]. Although vaccine development has focused largely on the endpoint of PA-targeted neutralising antibody, this alone is unlikely to confer sterilising immunity. At a general level, the CoP for effective AVA-vaccine-induced protection of macaques from anthrax challenge are IgG titre and IFNy+ T cell frequency against PA [43]. Protection conferred by anthrax spores is entirely dependent on CD4+ T cells [44].

In seeking an improved understanding of the interaction between *B. anthracis* and protection by the human immune system, a key question has been the impact of immunogenetic heterogeneity at the population level [45]. Work in mouse models has suggested that, as expected, both MHC and non-MHC polymorphisms influence these factors [46]. With respect to human vaccination, there is evidence of reduced immune responsiveness to PA in individuals with the DRB1*1501/DQB1*0602 haplotype [47]. In light of the importance of anti-PA immunity for protection and the relatively high frequency of this haplotype in many human populations, there is cause for concern in relation to vaccine efficacy and vaccine confidence. The situation is reminiscent of hepatitis B virus and MMR vaccinations, both of which demonstrate the profound influence of HLA polymorphism [48,49].

Our aim here has been to shed light on the role of HLA class II alleles in PA epitope presentation to the immune system and, thus, on disease outcome after anthrax challenge. A key experiment in this regard was to compare the impact of STI challenge on survival and the control of bacterial load in mice, all on a C57BL/6 background and lacking expression of endogenous murine MHC class II heterodimers, but differing in expression of specific HLA-DR or HLA-DQ alleles. The background C57BL/6 strain is considered one that mounts a low antibody response to anthrax PA and LF [46]. We found that HLA-DR15 transgenics (expressing the HLA-DRB1*1501 allele) were the most susceptible to challenge, echoing the results of human AVA HLA-DRB1*1501+ vaccinees [47]. It is particularly noteworthy that the effects of HLA class II alleles must be differentially effective in CD4+ T cell-mediated control of bacterial dissemination during the first 4 to 6 days after challenge, the very earliest days of detectable priming of an adaptive immune response. Nuanced differences in the potency and frequency of the initial CD4+ T cell responses have the potential to favourably impact survival by driving cellular responses to intracellular infection and generation of an initial neutralising antibody response. Such differences in susceptibility due to HLA polymorphisms are unlikely to have imposed evolutionary selection pressure in anthrax-exposed human populations. The pathogen is rarely transmitted from human-to-human, outbreaks tend to be of a limited nature (such as a local community consuming the same contaminated livestock), and most cases are not fatal. The greater concern relates to potential gaps in the efficacy of large-scale vaccination programmes for biodefense purposes, such as in the US military.

We looked at mechanisms underpinning HLA differences in susceptibility, starting with mapping of CD4+ T cell epitopes from PA. We previously described T cell memory responses to anthrax antigens in a cohort of individuals who suffered clinical disease after natural, occupational exposure [26,36]. These were agricultural workers from the Kayseri region of Turkey who had been in contact with infected livestock and been hospitalised with confirmed cutaneous anthrax infections. PBMCs were collected for immune analysis at 0.4 to 7.5 years after recovery under antibiotic therapy. Our key findings in this study were that natural infection elicits a considerably broader CD4+ epitope response than AVP vaccination and, at least in the setting of natural infection, PA is a very epitope-rich sequence, with epitopes spanning the entire length of the protein. It is well-established that in communities where environmental exposure to anthrax is relatively common, such as among goat-herders, exposure confers lifelong protection from re-infection [26]. Differences in antigen processing and generation of epitopes for HLA class II binding between the AVP subunit vaccine components and live infection of APC might, in some respects, have been predictable, except that earlier studies of dendritic cells treated with a lethal toxin showed a complete loss of the ability to effectively stimulate peptide-specific CD4+ T cells [50]. The PA sequence contained a number of regions with potential broad-ranging immunogenicity in terms of high-affinity binding to the majority of HLA class II alleles tested: 5 of the PA peptides analysed were relatively unusual in their capacity to bind very diverse HLA class II heterodimers at high affinity; PA 191–210, 331–350, 481–500, 591–610 and 711–730. The 191–200 PA epitope overlapped one that we have previously identified at the CD4+ T cell level as being strongly recognised in the memory T cell response of a 60-year old intravenous drug-user who survived injection of anthrax-contaminated heroin [28]. This collection of epitopes would be excellent candidates for a highly immunogenic, widely applicable, epitope-string vaccine. Importantly, the fact that all bound HLA-DRB1*1501 with high or very high affinity makes it likely that the ‘low-responder’ status of HLA-DRB1*1501 vaccinees would be overcome by an approach focused on these epitopes. However, HLA class II-related differences in susceptibility to anthrax challenge cannot be a simple question of relative availability of high-affinity HLA class II-binding PA epitopes to activate the CD4+ T cell repertoire. The most susceptible HLA allele that we identified, HLA-DRB1*1501, can present at least as many PA epitopes as can the least susceptible allele, HLA-DRB1*0101. It is also important to stress that, while the HLA transgenic mice used to define immunodominant PA epitopes, offer a useful reductionist system, the immune responses seen in this system may not fully recapitulate the effect of the individual HLA polymorphisms in a complete immune system. This may give a partial explanation for the divergence in epitopes identified in the HLA transgenics and those found in the human cohorts.

In summary, we draw two important conclusions from this comprehensive analysis of T cell recognition of anthrax PA. The first is that PA is an unexpectedly epitope-rich antigen, whether considered from the perspective of HLA class II binding or that of CD4+ T cell recognition. There appears to be a variable immunodominance hierarchy, which is dependent on both HLA class II genotype, and the context within which the immune system encounters the bacterial antigen (vaccination versus infection). The second key point, and one that offers an important note of caution to vaccinologists and to those planning biodefense strategies, is that there are likely to be major differences in both vaccine efficacy and anthrax severity imposed by HLA polymorphism within the population. These factors underscore the importance of considering immunological and vaccination strategies that can overcome such differences.

## Figures and Tables

**Figure 1 vaccines-10-01571-f001:**
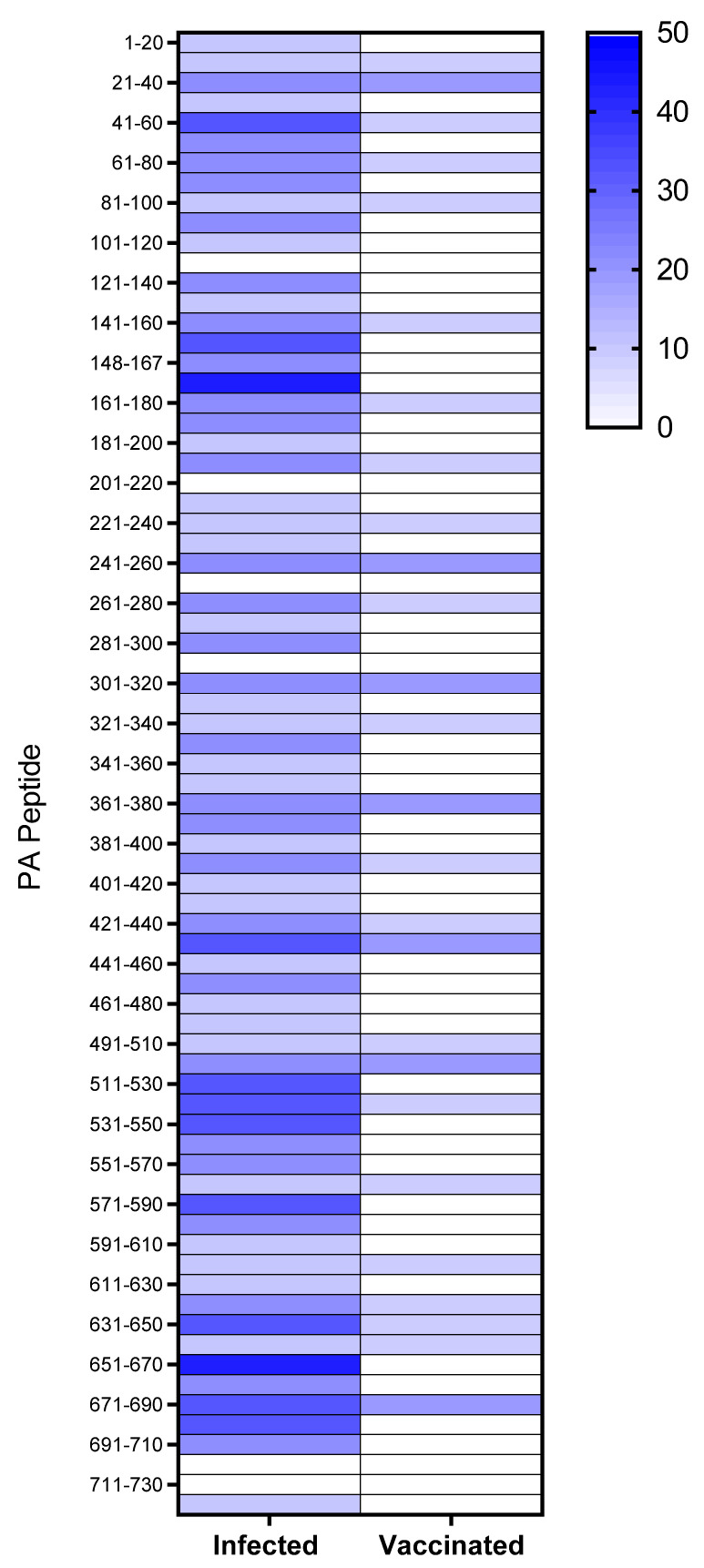
Heat map of CD4+ T cell epitope responses to anthrax PA domain I–IV peptides in human donors. Heat map representation of the epitope mapping results observed for positive CD4+ T cell IFNγ ELIspot responses in the human donor cohorts, comprising a total of 9 donors in the cutaneous anthrax (Kayseri) group and 10 donors in the AVP vaccinees (UK) group. Peptides were considered positive for the carriage of a CD4+ T cell epitope if the response was >50 SFC/106 PBMCs and 2SD above negative control, and the stimulation index (peptide response/negative control response) value was ≥1.5. The domains were defined as described previously; domain 1 = PA 1–20 to PA 241–260; domain 2 = PA 251–270 to PA 471–490; domain 3 = PA 491–510 to PA 581–600; domain 4 = PA 591–610 to PA 716–735, with some peptides overlapping the boundaries between domains) [38]. The colour bar at the right indicates the percentage of donors responding to a given epitope, with shading from white (0%) to dark blue (50%).

**Figure 2 vaccines-10-01571-f002:**
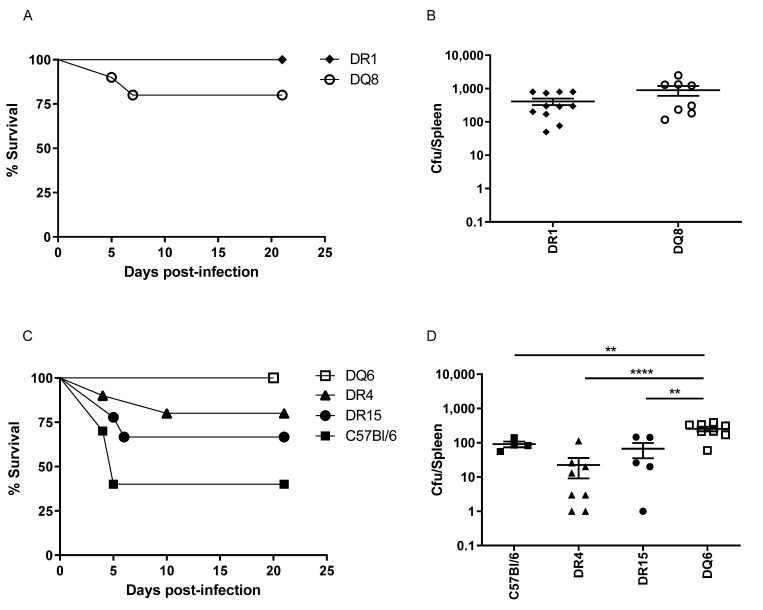
Differential susceptibility of HLA class II transgenic mice to anthrax infection. Groups of naïve HLA transgenic (DR1 n = 11, DQ8 n = 10, DR15 n = 9, DR4 n = 10, DQ6 n = 8) or C57Bl6 (n = 10) mice were challenged with either 10^5^ (**C**,**D**) or 10^6^ (**A**,**B**) CFU *B. anthracis* STI strain, in order to compare susceptibility. Mice were challenged intraperitoneally and their survival observed for 20 days post-infection. Percentage survival, together with mean splenic bacterial counts per HLA type, is shown for mice succumbing within the observation period (days 1 to 19) and for survivors culled at day 20. Statistical comparison of mean bacterial loads by mouse strain (**D**) indicated that higher bacterial loads were seen in DQ6 in comparison to; C57BL/6 (** *p* = 0.0093), DR4 (**** *p* < 0.0001) and DR15 (** *p* = 0.0014), (One-way ANOVA, Tukey’s multiple comparisons).

**Figure 3 vaccines-10-01571-f003:**
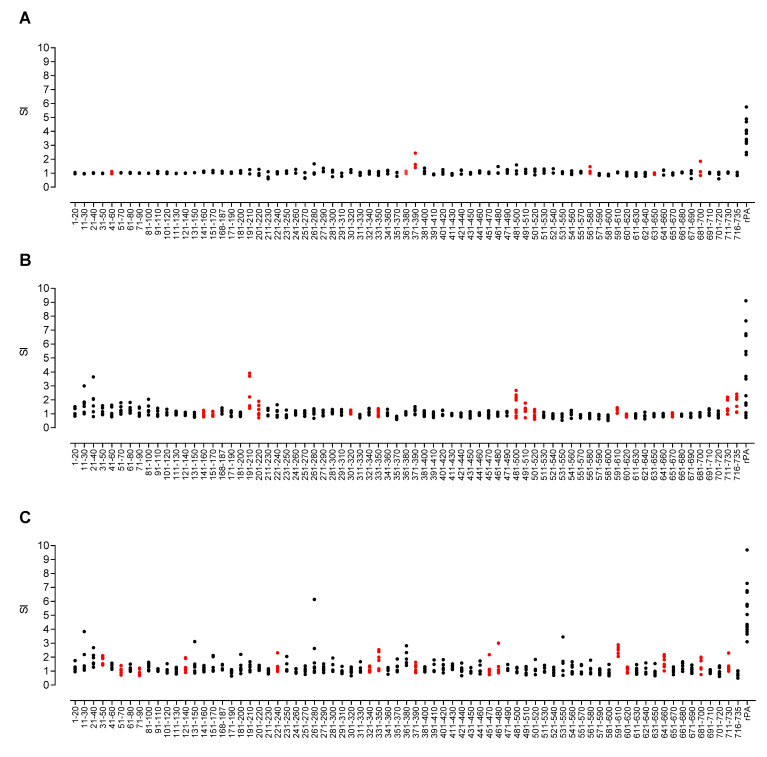
T-cell responses to PA peptides in whole rPA-immunised HLA-DR and HLA-DQ transgenic mice. Groups of HLA transgenic mice were immunised with the whole rPA protein in adjuvant, and the proliferative responses of draining lymph node cells to overlapping synthetic peptides representing the complete PA sequence were determined. Scatter plots show responses of individual mice transgenic for (**A**) HLA-DR1 (n = 3 for each peptide data point, and n = 11 for the rPA data point), (**B**) HLA-DQ8 (n = 6 for each peptide data point, n = 18 for the rPA data point) and (**C**) HLA-DR4 (n = 6 for each peptide data point, and n = 17 for the rPA data point). Data are presented as the SI calculated as the mean CPM of triplicate wells in the presence of peptide divided by the mean CPM in the absence of antigen. Values twice the mean CPM in the absence of antigen were considered positive responses. Confirmed epitopes are highlighted in red.

**Table 1 vaccines-10-01571-t001:** The PA peptides, identified in transgenic mouse strains and human cohorts, show relatively broad binding to common HLA-DR and HLA-DQ alleles.

PA Peptide Sequence	HLA Transgenic Strain Responding to Epitope after PA Immunisation	Human Cohort Responding to Epitope (>20% Cohort Responding)	Relative Binding of HLA Class II								
			DR1	DR3	DR4	DR7	DR11	DR13	DR15	DQ6	DQ8
^21^GYYFSDLNFQAPMVVTSSTT^40^	-	Vaccinee, Infected	23	60	0.3	22	53	>1908	118	ND	ND
^31^APMVVTSSTTGDLSIPSSEL^50^	DR4	-	ND	ND	ND	ND	ND	ND	ND	ND	ND
^41^GDLSIPSSELENIPSENQYF^60^	DR1	Infected	3312	>728	46	424	>1288	>1288	134	ND	ND
^51^ENIPSENQYFQSAIWSGFIK^70^	DR4	Infected	3	600	1	2	2	>2733	55	ND	ND
^61^QSAIWSGFIKVKKSDEYTFA^80^	-	Infected	617	12	650	89	14	4	1	ND	ND
^71^VKKSDEYTFATSADNHVTMW^90^	DR4	Infected	11	8	1	2	118	>2733	45	ND	ND
^91^VDDQEVINKASNSNKIRLEK^110^	-	Infected	1333	283	992	36	>1357	245	164	ND	ND
^121^QRENPTEKGLDFKLYWTDSQ^140^	DR4	Infected	>2563	800	6	1549	>1357	>2733	119	ND	ND
^141^NKKEVISSDNLQLPELKQKS^160^	DQ8	Infected	131	26	48	28	849	>2733	1	>3054	>166
^148^SDNLQLPELKQKSSNSRKKR^167^	-	Infected	ND	ND	ND	ND	ND	ND	ND	ND	ND
^151^LQLPELKQKSSNSRKKRSTS^170^	DQ8	Infected	>6667	>667	>1788	>1543	701	177	>511	>3054	>166
^161^SNSRKKRSTSAGPTVPDRDN^180^	-	Infected	>6667	>667	>1788	>1543	>1336	>1908	>511	ND	ND
^168^STSAGPTVPDRDNDGIPDSL^187^	-	Infected	>6667	>667	>1788	>1543	>1336	>1908	>511	ND	ND
^171^AGPTVPDRDNDGIPDSLEVE^190^	-	Infected	149	211	10	1167	7	51	95	ND	ND
^191^GYTVDVKNKRTFLSPWISNI^210^	DQ8	Infected	216	0.2	190	11	231	0.5	15	2488	>166
^201^TFLSPWISNIHEKKGLTKYK^220^	DQ8	-	3162	>667	1400	873	327	306	14	3077	>166
^221^SSPEKWSTASDPYSDFEKVT^240^	DR4	-	ND	ND	ND	ND	ND	ND	ND	ND	ND
^241^GRIDKNVSPEARHPLVAAYP^260^	DQ8	Vaccinee, Infected	2828	15	55	833	567	9	9	899	>166
^261^IVHVDMENIILSKNEDQSTQ^280^	-	Infected	>6667	>667	167	>1543	>1336	>1908	>511	ND	ND
^281^NTDSETRTISKNTSTSRTHT^300^	-	Infected	>6667	23	179	707	535	60	>511	ND	ND
^301^SEVHGNAEVHASFFDIGGSV^320^	DQ8	Vaccinee, Infected	>6667	>667	>1788	327	>1336	>1908	>511	3	7
^321^SAGFSNSNSSTVAIDHSLSL^340^	DR4	Infected	279	1	6	0.4	935	>1908	95	ND	ND
^331^TVAIDHSLSLAGERTWAETM^350^	DR4, DQ8	Infected	176	0.3	10	11	30	4	120	1056	0.1
^361^NANIRYVNTGTAPIYNVLPT^380^	DR1	Vaccinee, Infected	15	>728	0.4	1	12	>1288	3	ND	ND
^371^TAPIYNVLPTTSLVLGKNQT^390^	DR1, DR4	Infected	1	12	2	0.4	78	300	6	ND	ND
^391^LATIKAKENQLSQILAPNNY^410^	-	Infected	89	176	7	179	46	43	0.2	ND	ND
^421^LNAQDDFSSTPITMNYNQFL^440^	-	Infected	>6667	>667	1265	22	>1336	>1908	77	ND	ND
^431^PITMNYNQFLELEKTKQLRL^450^	-	Vaccinee, Infected	15	25	38	1	2	7	0.1	ND	ND
^451^DTDQVYGNIATYNFENGRVR^470^	DR4	Infected	200	>667	293	30	>1336	>1908	2	ND	ND
^461^TYNFENGRVRVDTGSNWSEV^480^	DR4	-	ND	ND	ND	ND	ND	ND	ND	ND	ND
^481^LPQIQETTARIIFNGKDLNL^500^	DQ8	Infected	2	5	886	0.3	37	2	1	240	6
^491^IIFNGKDLNLVERRIAAVNP^510^	DQ8	-	3801	31	327	267	0.1	7	95	693	75
^501^VERRIAAVNPSDPLETTKPD^520^	DQ8	Vaccinee, Infected	721	75	69	55	>1288	>1288	145	2506	29
^511^SDPLETTKPDMTLKEALKIA^530^	-	Infected	211	10	1800	401	1000	10	37	ND	ND
^521^MTLKEALKIAFGFNEPNGNL^540^	-	Infected	1155	18	>1788	98	189	10	77	ND	ND
^531^FGFNEPNGNLQYQGKDITEF^550^	-	Infected	>6667	>667	207	>1543	1134	>1908	63	ND	ND
^541^QYQGKDITEFDFNFDQQTSQ^560^	-	Infected	>2563	25	306	>3365	>1357	>2733	44	ND	ND
^551^DFNFDQQTSQNIKNQLAELN^570^	-	Infected	249	82	239	267	>1336	250	200	ND	ND
^561^NIKNQLAELNATNIYTVLDK^580^	DR1	-	2	>728	76	8	137	64	8	ND	ND
^571^ATNIYTVLDKIKLNAKMNIL^590^	-	Infected	31	5	278	19	0.5	3	11	ND	ND
^581^IKLNAKMNILIRDKRFHYDR^600^	-	Infected	2160	0.1	500	378	6	0	4	ND	ND
^591^IRDKRFHYDRNNIAVGADES^610^	DR4, DQ8	-	25	1	0.2	4	4	25	2	1132	0.5
^601^NNIAVGADESVVKEAHREVI^620^	DR4, DQ8	-	4989	10	1183	750	732	16	122	2191	6
^621^NSSTEGLLLNIDKDIRKILS^640^	-	Infected	2236	1	414	80	53	3	77	ND	ND
^631^IDKDIRKILSGYIVEIEDTE^650^	DR1	Infected	5	775	510	1	72	1026	0.3	ND	ND
^641^GYIVEIEDTEGLKEVINDRY^660^	DR4	-	3162	82	21	65	433	1333	0.1	ND	ND
^651^GLKEVINDRYDMLNISSLRQ^670^	DQ8	Infected	7	10	6	22	46	15	6	>3054	1
^661^DMLNISSLRQDGKTFIDFKK^680^	-	Infected	200	2	30	27	33	4	27	ND	ND
^671^DGKTFIDFKKYNDKLPLYIS^690^	-	Vaccinee, Infected	2494	100	>1788	138	14	25	4	ND	ND
^681^YNDKLPLYISNPNYKVNVYA^700^	DR1, DR4	Infected	1	4	1	2	2	30	1	ND	ND
^691^NPNYKVNVYAVTKENTIINP^710^	-	Infected	50	3	75	0.3	13	6	10	ND	ND
^711^SENGDTSTNGIKKILIFSKK^730^	DR4, DQ8	-	>6667	>667	>1788	133	3	217	3	>3054	>166
^716^TSTNGIKKILIFSKKGYEIG^735^	DQ8	-	183	1	849	35	0.4	0.4	1	>3054	>166

Binding affinities are expressed as relative values which were calculated as the ratio of the PA peptide IC50 to the IC50 of a reference peptide chosen as a high binder for each allele. Good binding affinity values were interpreted as <100. Means were calculated from at least three independent experiments. ND = Not Done.

## Data Availability

The data presented in this study are available on request from the corresponding authors.

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
