# Peer review of "Impact of HLA Polymorphism on the Immune Response to Bacillus Anthracis Protective Antigen in Vaccination versus Natural Infection"

_vaccines, 2022, doi:10.3390/vaccines10101571_

Round 1
Reviewer 1 Report
The authors present the relevance of different HLA class II in transgenic mice when exposed to Protective Antigen of Bacillus anthracis. The article is in general well and clearly written.
Only minor improvements are suggested by the reviewer.
1. Maybe a bit of background about HLA and why is it relevant during B. anthracis infection could be helpful.
2. In line 397 the reference should be adjusted to the legend size
3. Could an ELISPOT also bring information to the study?
4. It would be interesting to discuss why intranasal infection is not used to challenge the mice since that could mimic one important route of exposure that takes place in humans
Author Response
Reviewers comments
Reviewer 1
The authors present the relevance of different HLA class II in transgenic mice when exposed to Protective Antigen of Bacillus anthracis. The article is in general well and clearly written.
Only minor improvements are suggested by the reviewer.
- Maybe a bit of background about HLA and why is it relevant during B. anthracis infection could be helpful.
Within the introduction, we have expanded on the background regarding HLA presentation of B. anthracis antigens to CD4+ T cells and their role in bacterial infection;
‘Recent advances in vaccinology and microbial immunology have indicated that there is strong evidence for HLA polymorphisms as a determinant of susceptibility and resistance to disease, and previous work from our lab has shown that individuals naturally exposed to anthrax spores demonstrate IFNγ secreting antigen-specific CD4+ T cell immunity to PA and LF, which for PA, showed correlation between the magnitude of response and the duration of the infection [26,27]. We also found that a survivor of injectional anthrax developed strong, potentially protective, T cell immunity to several commonly immunodominant epitopes of PA and LF, previously described in Turkish patients [28]. This evidence suggests that cellular immunity, especially mediated by antigen presentation through both HLA-DR and DQ alleles to a specific TCR repertoire, has a critical role to play in vaccine mediated clearance of B. anthracis.’
- In line 397 the reference should be adjusted to the legend size
The reference has been adjusted to the legend size.
- Could an ELISPOT also bring information to the study?
We agree with the reviewer that IFNy ELISpots could provide extra information regarding the CD4+ response in the mice, however it was not logistically possible (or ethically acceptable, in terms of increased animal usage) to map the T cell epitopes in the HLA transgenic mice using both proliferative responses (the main readout in this study) and ELISpots. We anticipate that this manuscript will provide valuable information, regarding PA epitopes, to the field of bacterial immunology and stimulate research further investigating Th1/2/17 cytokine release using such assays.
- It would be interesting to discuss why intranasal infection is not used to challenge the mice since that could mimic one important route of exposure that takes place in humans
This study expressly focuses on mapping the epitopes related to cutaneous anthrax infection and AVP vaccination in humans. We used the HLA class II transgenic mice as a defined, reductionist model to allow more detailed analysis of individual alleles in isolation and compared these to the human responses; we therefore chose a well-established route of infection in the mouse model which would allow us to mimic the human exposure route more closely. Furthermore, while mice are susceptible to the attenuated strains of B.anthracis (such as STI, which is a Russian human vaccine strain) used in such infection experiments, the pathology associated with a respiratory or intranasal route of infection is different to that seen in the human lung, due primarily to differences in the way that alveolar macrophages respond. Therefore, it would also be important in any such studies to use more suitable animal models (eg rabbits or primates) to fully recapitulate human inhalational anthrax. We agree with the reviewer that exploring the immune response to bacterial antigens following inhalational anthrax infection would be an interesting avenue of research, however it is outside the scope of this study.
Reviewer 2 Report
This is an interesting article by Stephanie Ascough et al.
I would like to address a small number of suggestions to you that may improve the manuscript.
Abstract and Introduction
Lines 24, 44, 56, 100, 103 - please write Bacillus anthracis in italics.
Please clearly write the aim of present study.
Materials and methods
Inclusion and exclusion criteria must have to be included.
Results
Results section must be started with results from the present study but not from your previous data.
Please remove first paragraph (lines 260-269) to discussion section.
Please analytically report your data from figure 2 in the text in section "Differential susceptibility to B. anthracis challenge in HLA transgenic mice".
Figure 2 A is represented in sentence "The groups challenged with 106 CFU (DR1, DQ8) had high bacterial loads," isn’t it? (Line 322)
Conclusions are too short.
